# Mid-Infrared Tunable Laser-Based Broadband Fingerprint Absorption Spectroscopy for Trace Gas Sensing: A Review

**Zhenhui Du [1,2,\*]** , **Shuai Zhang [1]** , **Jinyi Li [3]** , **Nan Gao [2,4]** and **Kebin Tong [1]**

[1] State Key Laboratory of Precision Measuring Technology and Instruments, Tianjin University, Tianjin 300072, China; Shuai_zhang@tju.edu.cn (S.Z.); tongkebin@tju.edu.cn (K.T.)

[2] Key Laboratory of Micro Opto-electro Mechanical System Technology, Tianjin University, Ministry of Education, Tianjin 300072, China; ngao@hebut.edu.cn

[3] Key Laboratory of Advanced Electrical Engineering and Energy Technology, Tianjin Polytechnic University, Tianjin 300387, China; lijinyi@tjpu.edu.cn

[4] School of Mechanical Engineering, Hebei University of Technology, Tianjin 300130, China

\* Correspondence: duzhenhui@tju.edu.cn; Tel.: +86-138-205-95185

**Abstract:** The vast majority of gaseous chemical substances exhibit fundamental rovibrational absorption bands in the mid-infrared spectral region (2.5–25 μm), and the absorption of light by these fundamental bands provides a nearly universal means for their detection. A main feature of optical techniques is the non-intrusive in situ detection of trace gases. We reviewed primarily mid-infrared tunable laser-based broadband absorption spectroscopy for trace gas detection, focusing on 2008–2018. The scope of this paper is to discuss recent developments of system configuration, tunable lasers, detectors, broadband spectroscopic techniques, and their applications for sensitive, selective, and quantitative trace gas detection.

**Keywords:** tunable laser absorption spectroscopy; mid-infrared fingerprint spectrum; broadband spectrum; trace gas detection; wavelength modulation spectroscopy; quantum cascade lasers; interband cascade lasers

## 1. Introduction

Laser-based trace gas sensing is becoming more popular in a wide variety of areas including urban and industrial emission measurement [1,2], environmental and pollution monitoring [3,4], chemical analysis and industrial process control [5,6], medical diagnostics [7,8], homeland security [9], and scientific research [10,11]. With the increase in global environmental, ecological, and energy issues, laser-based trace gas detection technology has attracted unprecedented attention. Laser gas sensing is based on the analysis of characteristic spectra of molecules, mostly known as tunable diode laser absorption spectroscopy (TDLAS) or tunable laser absorption spectroscopy (TLAS). Laser absorption spectrometers (LAS), also known laser gas analyzers (LGA), enjoy the merits of non-contact, fast response time, high sensitivity and selectivity, the potential to be calibration-free, low maintenance requirements, and a long life cycle. LAS is particularly suitable for in situ, online analysis, and real-time detection.

Laser gas sensing has undergone tremendous progress along with the advancement in tunable semiconductor lasers in the last few decades. Direct absorption spectroscopy (DAS) is the most common technique for simple optical configuration, signal processing, and potential absolute measurement. DAS often suffers from low sensitivity (absorbance ~$10^{-3}$) due to the interference from $1/f$ noise in the system and laser power fluctuation. There are basically two ways to improve the sensitivity

in this situation: (1) reduce the noise in the signal; or (2) increase the absorption. The former can be achieved by using modulation technique, e.g., wavelength modulation spectroscopy (WMS) and frequency modulation spectroscopy (FMS), with a typical sensitivity of absorbance $\sim 10^{-5}$. The latter can be obtained by placing the gas inside a cavity in which the light passes through multiple times to increase the interaction length, e.g., multiple-pass or long path absorption cells, and cavity enhanced absorption spectroscopy (CEAS) [12,13]. Both ways of reducing noise and increasing absorption can be applied to a same system, e.g., cavity enhanced wavelength modulation spectrometry [13] and noise-immune cavity-enhanced optical heterodyne molecular spectroscopy (NICE-OHMS) [14,15].

FMS is a method of optical heterodyne spectroscopy capable of rapid measurement of the absorption or dispersion associated with narrow spectral features. The absorption or dispersion is measured by detecting the heterodyne beat signal that occurs when the FMS optical spectrum of the probe wave is distorted by the spectral feature of interest. Recently, dispersion spectroscopy, namely chirped laser dispersion spectroscopy (CLaDS) [16] or heterodyne phase sensitive dispersion spectroscopy (HPSDS) [17], has attracted attention for its immunity to optical intensity changes and superb linearity in the measurement of concentration.

CEAS and its new versions, e.g., cavity ring-down spectroscopy (CRDS) [18], broadband cavity ring-down spectroscopy [19], phase-shift cavity ring-down spectroscopy [20], integrated cavity output spectroscopy (ICOS) [21], and continuous wave cavity enhanced absorption spectrometry (cw-CEAS) [22], provide much larger pathlength enhancement by using a resonant cavity, and thus have highly sensitive absorbance $\sim 10^{-7}$–$10^{-9}$.

Practically, the simplest and most promising method to enhance the signal of trace gas detection is to perform the detection at wavelengths where the transitions have larger line strengths, e.g., using fundamental rovibrational bands or electronic transitions. The fundamental rovibrational bands of a vast majority of gaseous chemical substances, located at the mid-infrared spectral region (MIR, 2.5–25 μm), are due to the transitions of molecular rovibrational energy states. In general, these bands have stronger line strengths than the overtone and combination bands typically used in the visible and near-IR regions. The MIR spectrum depends on the physical properties of the molecule such as the number and type of atoms, the bond angles, and the bond strength. Thus, the MIR spectrum is uniquely characterized by highly specific spectroscopic features and is considered the molecular signature, which allows both the identification and quantification of the molecular species, especially suitable for larger molecules, e.g., volatile organic compounds (VOCs) [23].

VOCs are gaseous organic chemicals at the conditions of normal temperature and pressure (NTP, 293.15 K and 101.325 kPa). There are several hundred types of VOCs, some of which are dangerous to human health or cause harm to the environment. VOCs monitoring has attracted attention for long time. Commonly used spectroscopic techniques for VOCs detection are Fourier-transform infrared spectroscopy (FTIR) and differential optical absorption spectroscopy (DOAS) [24]; however, their sensitivity, selectivity, and fragile optical setup are not always sufficient for harsh applications.

Recently, newly commercialized MIR detectors and lasers, especially quantum cascade lasers (QCLs) [25] and interband cascade lasers (ICLs) [26], have stimulated the development of high-performance, compact, and rugged gas sensors. Traditionally, TLAS use a discrete narrow absorption lines of small molecules for gas sensing. For larger molecules, however, so many lines overlapping with each other results in the spectral features being broad and smooth, except for occasional spikes [23,27]. These spectral features are distinct from those of the discrete narrow absorption lines with a Lorentzian, Gaussian, or Voigt profile. Detection of trace gas with broadband absorption is much more difficult than with an isolated narrow spectral line. Extra effort should be made to cope with the challenges of the broadband of larger molecules.

In this paper, we primarily review tunable laser-based broadband absorption spectroscopy for trace gas detection in 2008–2018. After a brief overview of the principle (Section 2), we discuss the system configuration, including MIR tunable lasers, detectors, and optical configuration in Section 3. We discuss broadband spectroscopic techniques concerning derivative spectroscopy (Section 4), WMS

(Section 5), and optical frequency comb spectroscopy (Section 6). Section 7 is a collection of MIR gas sensing applications. Section 8 gives conclusions and prospects.

## 2. Principle

Quantitative spectral analysis is based on the Beer-Lambert law, which gives the relationship between the incident and the transmitted radiation through a gas cell filled with a molecular gas sample:

$$I(v) = I_0(v) \times exp\{-\sigma(v) \times L \times C\}, \tag{1}$$

where $I_0$ and $I$ are the incident and transmitted radiant powers, respectively; $\sigma$ is absorption cross section of the molecule in $cm^2/molecule$; $L$ is absorption pathlength in cm; $C$ is the density of the molecule in $molecule/cm^3$. Usually, the absorption cross section $\sigma$ is also used to describe the absorption intensity. The line strength is retrieved by spectrally integrating the absorption line shape and applying the ideal gas law:

$$S(T) = \frac{K_B TA}{X_i LP \, r_{iso}}, \tag{2}$$

where $K_B$, $T$ (K), and $P$ (Pa) are the Boltzmann constant, gas temperature, and total pressure of the gas sample, respectively; $X_i$ is the amount fraction of $i$ species; $A$ ($cm^{-1}$) is integral absorbance; $r_{iso}$ is a correction factor for isotopic fractionation of the gas sample.

WMS is commonly used to improve the sensitivity of gas sensing. The WMS theory and signal model have been detailed previously [28], and so are only briefly reviewed here. A periodic sawtooth ramp ridden by a high-frequency sinusoidal is applied to the laser injection current, thus the laser wavenumber $v(t) = v_c + v_a \cos \omega t$ is scanned across the transition of gas to be detected, where $v_c$ and $v_\alpha$, are the laser center wavenumber and modulation depth, respectively; $\omega$ is the radian frequency. In case of ideal conditions, ignoring all kinds of interference, the modulated absorption signal is detected by a photodiode and then processed using a lock-in amplifier to demodulate the signal at the harmonics (1*f*, 2*f*, 3*f*, etc.). The second harmonic component (WMS-2f) is commonly used for calculating the concentration of target gas. In the case of optically thin ($\sigma(v) \cdot L \cdot C \leq 0.05$), the ideal 2*f* signal is modeled as:

$$A_{ideal \, 2f} = \frac{2I_0 CL}{\pi} \int_0^\pi -\alpha(v_c + v_a cos\theta)cos2\theta d\theta \propto I_0 CL, \tag{3}$$

where $\alpha$ is the absorption coefficient and $\theta = \omega t$ is the phase angle. When the incident laser intensity $I_0$ and optical path $L$ are constant, the amplitude of WMS-2f signal is proportional to the gas concentration. Practically, apart from the 2*f* signal descript in Equation (3), the detected signal consists of random noise and the derivation of optical fringes [29,30]. The optical fringes appear as unpleasant spectral features that are usually mixed with the target absorption, and constitute one of the major obstacles in the gas detection. In a well-designed and -fabricated system, the optical fringes should be reduced, and only small residual fringes remain with sinusoidal waveforms, while random noise is seen as small time-varying wiggles superimposed on the true underlying signal, with small standard deviation. Thus, the detected signal could be described as:

$$A_{\text{detected } 2f} = e_n + \sum a_j(t) \times \cos(\omega_j(t) \times t) + A_{\text{ideal } 2f}, \tag{4}$$

where $\alpha_j(t)$ and $\omega_j(t)$ are the instantaneous amplitude and frequency of *j*th fringe component, respectively; $A_{\text{ideal}} 2f$ is the WMS-2f signal modeled by Equation (3). The profiles of second harmonic of absorption, fringes, and noise will inherit the featuresof their origination. These profile differences among WMS-2f, harmonic of optical fringes, and noise will be a novel breakthrough point to distinguish and eliminate the interference from the signal (details in Section 5.3).

## 3. System Configuration

A typical LAS consists of a laser, a photodetector, and an optical configuration for light interaction with gas. For WMS-based LAS, there are, additionally, a laser modulator and a signal demodulator, the latter usually by a lock-in amplifier (LIA).

The laser is the LAS's key component; it usually needs to be continuously tunable mode-hop-free, reliable, single frequency with narrow linewidth (typically <1 MHz), and to have low noise intensity. Historically, lead-salt diode lasers have been developed in a MIR gas sensor. However, these lasers require cooling to liquid nitrogen temperatures and present problems for mode hops and multi-mode operation. Recently, great progress in laser technology has brought many types of excellent lasers, e.g., QCLs and ICLs.

High-sensitive and low-noise detectors are essential for trace gas detection. The most popular commercial infrared detector is a mercury-cadmium-telluride (MCT, or HgCdTe) photoconductive semiconductor-based detector. The MCT detector enjoys a very wide spectral response (2 to 25 μm) and higher speed of detection. Its main limitation is that it needs cooling to reduce noise due to the thermally excited current carriers. Alternatively, newly developed quantum heterostructure detectors could play a vital role in future infrared detection [31].

The optical configuration provides interaction between light and gas' the interaction length directly relates with the sensitivity of gas detection. Thus, a long interaction length is desired to achieve high sensitivity. Multiple-pass cells (MPCs) and open long path are commonly used in LAS to measure low-concentration components or to observe weak spectra in gas. The requirements of compactness, small sample volume, and fast response time stimulated the development of a new type of gas cell. Recently, a hollow waveguide (HWG)-based gas cell has been found to boast the advantages of small sample volume and fast response time [8,32], whereas substrate-integrated hollow waveguides (iHWG) are compact integrated sensors [33]. On the other hand, the need for non-fixed open-path gas detection, e.g., leak detection, aroused the development of standoff remote sensing without a retroreflector [34–36].

### 3.1. Mid-Infrared Tunable Lasers

MIR tunable laser technology has undergone tremendous development in the past decade, which involves QCL, ICL, difference frequency generator (DFG) [37], optical parametric oscillator (OPO) [38], fluoride fiber lasers [39], hollow-core fiber gas laser [40], VCSEL [41], and II–VI chalcogenides-based MIR lasers [42]. As there have been many excellent reviews of MIR light sources [25,43–45], here we present only a brief overview of the newly developed, highly reliable, and most widely used in spectroscopy, focusing on QCL, ICL, VCSEL, and optical frequency comb (OFC).

### 3.1.1. Quantum Cascade Lasers

QCL, first demonstrated by Faist et al. in 1994 [46], emits by intersubband transitions between energy levels inside superlattice quantum wells rather than the material bandgap energies in conventional lasers. The most attractive distributed feedback (DFB) QCLs were commercialized in 2004. Now, many commercial providers offer cw- and RT-operated QCLs in different configurations ranging from Fabry-Perot devices, to DFB resonators, to external cavities-based (EC-) QCLs, as well as high-power devices (with watts) in the infrared to terahertz spectral region. QCLs are attractive for infrared countermeasure, metrology, high-resolution spectroscopy, and chemical sensing applications [7,47–53].

QCLs enjoy a broad gain profile of hundreds of wavenumbers and a narrow linewidth about 0.5 MHz by providing monochromatic feedback, i.e., DFB and EC [54]. The natural linewidth can be as low as a few hundred hertz. The emission wavelength of DFB-QCLs can be tuned only a couple of wavenumbers, whereas EC-QCLs can provide hundreds of wavenumbers of coverage with drawbacks of slow mechanical speed, instability, alignment of multiple optical components, and high

price. Broadband QCLs have been also built by an array of DFB QCLs with closely spaced emission wavelengths and fabricated monolithically with wavelength coverage of several micrometers [55]. QCL arrays enjoy broad tuning and high spectral resolution, which opens up a wide range of new possibilities for fast, compact, and mechanically robust solutions with high customizability [56].

Overall, QCLs have been greatly improved, but many challenges remain, such as intervalley scattering, heat removal from the core region, and interface scattering, which limit the performance of QCLs, especially at short wavelengths [57]. Although InP-based QCLs emitting at wavelengths of 3–4 µm have been demonstrated recently [58,59], they are still not commercially available.

### 3.1.2. Interband Cascade Lasers

ICL was presented by Yang in 1995 [60]. RT-cw operation of ICL was first demonstrated in 2008 [61]. Only a few years ago, DFB-ICLs became commercially available with an optical power of milliwatts and a spectral tuning range of a few wavenumbers by Nanoplus GmbH [26]. Like QCLs, ICLs employ the concept of bandstructure engineering to achieve an optimized laser design and reuse injected electrons to emit multiple photons. However, ICLs' photons are generated by interband transitions rather than the intersubband transitions used in QCLs, which allows ICLs to achieve lower input powers than is possible with QCLs.

DFB-ICLs provide single mode, narrow linewidth (~0.7 MHz) [62], low power consumption (hundreds of milliwatts), a compact system solution, and RT-cw emission in the 3–6 µm range, which fill the MIR gap perfectly. DFB-ICLs have been used for the ppb-level detection of many important gases, especially hydrocarbon species [63–69], which leads to many important applications in various areas, for example, clinical diagnostics [70,71], combustion probing [72], environmental monitoring [73,74], and remote sensing.

### 3.1.3. Mid-Infrared Vertical-Cavity Surface-Emitting Lasers

MIR VCSELs have been paid great attention in the past decade for their advantages of low power consumption, low beam divergence, narrow and single-fundamental-mode, high wavelength tunability, and on-wafer testing capability [45]. The buried tunnel junction (BTJ) concept yields high-performance VCSELs in the wavelength ranges of 1.3–2.6 µm [75] and 2.3–3.0 µm [76], respectively. An interband cascade VCSEL has achieved lasing to $\lambda$~3.4 µm in pulsed mode at temperatures up to 70 °C [77], and a 4 µm VCSEL has been developed by using a single-stage active region with eight type-II quantum wells combined with BTJ technology [78]. VCSELs operate with very low threshold currents of several mA and very low power consumption of milliwatts. They provide a single mode by distributed Bragg reflector with a moderate linewidth of tens of MHz. VCSELs are particularly suited for compact and battery-powered sensors.

### 3.1.4. Mid-Infrared Optical Frequency Comb

OFCs consist of a series of discrete, narrow, stable, equally spaced spectral lines that have a fixed phase relationship between them. These combs can span a broadband of frequency range that have found important applications in areas such as metrology, spectroscopy, and optical communications [79]. To date, OFC sources have extended from the ultraviolet, visible, infrared, and terahertz spectral regions [80]. We only focus on the MIR-OFC, including mode-locked, difference frequency generators (DFG), optical parametric oscillation (OPO), and direct modulation OFC.

**Mode-Locked Laser**. The most popular way of generating a frequency comb is with a mode-locked laser. Mode-locked lasers produce a series of optical pulses separated in time by the round-trip time of the laser cavity. The spectrum of such a pulse train approximates a series of Dirac delta functions separated by the repetition rate of the laser. In the past decade, there have been hundreds of demonstrations of generating OFC with various lasers including Ti: sapphire solid-state lasers [81], Er: fiber lasers [82], Kerr-lens mode-locked lasers [83], QCLs [84], and ICLs [85] with

repetition rates typically between to MHz to 10 GHz. The issues with mode-locked OFC are lower output power, still experimental, and not commercially available.

**Difference Frequency Generators**. DFG is the most versatile method to generate mode-hop-free tunable broad laser by a nonlinear optical process. The two necessary conditions to achieve MIR OFC output are high-coherence pump lasers and a suitable order nonlinear crystal. There are more than a dozen demonstrated crystals, among which LiNbO3 and ZnGeP2 are the most popular. DFG-based OFC have more power and stability and do not require an oscillating cavity or a high threshold of the optical parametric process [86], whereas the drawbacks are high cost and a relatively low conversion efficiency [37,86,87].

**Optical parametric oscillator**. An OPO is another parametric nonlinear optical process used to provide a high output power and versatile sources of coherent radiation for spectral regions inaccessible to lasers. OPO has been shown providing high power OFCs based on Yb: fiber laser covering several micrometers [88,89]. Researchers have not only shown that degenerate synchronously OPOs are efficient tools to transfer near-infrared (NIR) frequency combs to the mid-infrared; also, they can utilize pump lasers in NIR and expand the spectrum to the mid-infrared [38,90]. OPO offer high output power with broad spectral coverage, but require a free-space resonator, external pumping sources, and many optical components [91]. This makes OPO sources bulky, vulnerable to any external disturbances, and very impractical in field applications.

**Direct Frequency-Modulation Combs**. Direct FM combs have been experimentally demonstrated in semiconductor lasers, such as QCLs [91–97], ICLs [85], quantum dot (QD) [98], and quantum dash lasers [99]. These lasers are passively mode-locked with cw or quasi-cw output. Direct FM combs enjoy broadband, wide repetition frequency from kHz to THz [100] with line width of kHz to MHz level. Direct FM comb offers the possibility of a portable, chip-scale device with low power consumption, which are desired in spectroscopic trace gas sensing.

### 3.2. Infrared Detector

There has been exciting progress in MCT junction technology, which could design and fabricate a high-performance MCT photovoltaic detector for operation in RT and in situ applications [101]. The commercialized detector benefits from the use of optimized material, device architecture, concentrators of radiation, enhanced absorption, and shields against thermal radiation, and can achieve directivity as high as $10^{11}$ cm$\cdot\sqrt{\text{Hz}}\cdot\text{W}^{-1}$ in RT conditions [102]. Alternatively, more progress has been noted in quantum engineering-based detectors.

### 3.2.1. Quantum Heterostructure Detector

Quantum heterostructure-based infrared detectors, including quantum cascade (QC), quantum well (QW), quantum dot, and combinations of both QDs and QWs in a dot-in-a-well (DWELL) strategy detectors, have high detectivity and low dark currents [31,103,104]. QC detectors have low noise due to their low interference of background radiation. QW detectors are characterized by a narrow spectral band and are easily fabricated. However, the disadvantages of QW detectors include low operating temperatures and the requirement of a scattering filter to adjust the incident light angle [105]. QD detectors have higher operating temperatures and are capable of absorbing normally incident photons. However, they have much lower quantum efficiency due to the lower absorption and capture probabilities of incident light [104,106]. DWELL devices have shown promise in terms of detectivity and spectral range, though their operating temperatures remained fairly low.

Most quantum heterostructure detectors are still at the experimental stage, with different fabrication technologies and materials being used. Few commercial products have been reported, which leads to the restriction of such detectors' application in MIR detection. However, recent advances in this field such as new device-chip hybridization [107], antimonide-based membrane synthesis integration, and strain engineering [108] may be scalable methods for the fabrication of commercially available heterostructure detectors.

### 3.2.2. Infrared Avalanche Photodiodes

An avalanche photodiode (APD) provides an internal multiplication necessary to achieving high avalanche gain at low bias with low noise and high bandwidth [109]. Presently, MCT APD is the most promising for trace gases in standoff remote sensing with high sensitivity. The demands of night vision and LIDAR systems stimulate the development of MCT APD arrays [110]. Overall, commercialized high-sensitive and bandwidth MCT APD would play a vital role in many areas.

### 3.3. New Types of Gas Cells

Recently, new types of gas cells have been developed to achieve long optical path in a smaller volume, including modified MPCs [111–116], circular multi-reflection (CMR) cells [117–124], and HWG [32,63,69,125–128]. Guo and Sun reviewed the progress in modified MPCs and CMR cells and compared them in terms of optical pathlength (OPL), volume, and path-to-volume ratio (PVR) [116]. HWG-based gas cells have been discussed in detail in books and reviews [129–131]. We compared the main features among the modified MPCs, CMR cells and HWG, as shown in Table 1.

**Table 1.** Comparison of different types of gas cells.

| Type | | Ref. | OPL/m | Volume/L | PVR/m/L | ADs | DADs |
|---|---|---|---|---|---|---|---|
| Modified MPCs | 2 mirrors | [111] | 22 | 0.55 | 40 | (c) (d) (e) | ②⑤ |
| | 6 mirrors | [112] | 314 | 1.25 | 251.2 | (b) (c) (e) | ①②⑤ |
| | 3 mirrors | [113] | 1.46 | 0.33 | 4.4 | (c) (d) | ①②⑤ |
| | 2 mirrors | [114] | 57.6 | 0.225 | 256 | (a) (b) (d) (e) | ②⑤ |
| | 2 mirrors | [115] | 26.4 | 0.28 | 94.3 | (b) (c) (e) | ②⑤ |
| | 6 mirrors | [116] | 32.4 | 0.48 | 67.5 | (c) (d) (e) | ①②⑤ |
| CMR cells | 1 mirrors | [117] | 1.05 | 0.078 | 13.5 | (c) (d) (e) | ③④ |
| | 6 mirrors | [118] | 3.1 | 0.024 | 129.2 | (a) (c) (d) (e) | ③④ |
| | 1 mirrors | [119] | 2.16 | 0.04 | 54 | (a) (c) (d) (e) | ③④ |
| | 1 mirrors | [120] | 4.08 | 0.04 | 102 | (a) (c) (d) (e) | ③④ |
| | 6 mirrors | [121] | 0.69 | 0.013 | 53.1 | (a) (c) (d) (e) | ③④ |
| | 1 mirrors | [122] | 12.24 | / | / | (a) (c) (d) (e) | ③④ |
| | 1 mirrors | [123] | 9.9 | 0.05 | 198 | (a) (c) (d) (e) | ③ |
| | 65 mirrors | [124] | 10 | 0.14 | 71.4 | (a) (c) (d) (e) | ③④ |
| HWGs | Ag/AgI-HWG [32,69] | | 5 | 0.004 | 1250 | (a) (d) (e) | ④ |
| | iHWG | [126] | 0.25 | 0.001 | 250 | (a) (d) (e) | ④ |
| | iHWG | [63] | 0.075 | 0.0003 | 250 | (a) (d) (e) | ④ |
| | PBF-HWG | [127] | 1 | $5 \times 10^{-6}$ | 200,000 | (a) (d) (e) | ④⑤ |
| | PBF-HWG | [128] | 0.08 | $4.5 \times 10^{-7}$ | 177,778 | (a) (d) (e) | ④⑤ |
| Advantages index | | | Disadvantage index | | | | |
| (a) easy configurations, fewer than three components (b) ultra-long pathlength (>50 m) (c) keeping focal properties, low-aberrations (d) compact structure (e) suitable for MIR | | | ① multiple components (≥3 optical elements) and requiring adjustment ② large volume (>0.1 L) ③ hard to align, sensitive to vibrations or input conditions ④ spot diffusion introduced by aberrations ⑤ slow gas exchanging | | | | |

There are three categories of HWG employed in spectroscopic gas sensing, namely Ag/AgI-coated HWG (Ag/AgI-HWG), photonic bandgaps HWG (PBG-HWG), and iHWG [131], and the comparison

of them is as shown in Table 1. HWGs work as both an optical waveguide and gas transmission cell that provide an extended OPL yielding high sensitivity measurements [69]. It is worth noting that filling PBG-HWG with analyte gas for sensing is difficult owing to the considerable back-pressure building up in the hollow structure. HWGs are ideal candidates for gas cells due to their high PVR and ability to transmit light at a fairly wide wavelength range, which means building a MIR sensor platform is feasible.

### 3.4. Fully Integrated Sensors

Profiting from the development of MIR lasers, detectors, and gas cells, as described above, the laser spectrometer can be potentially integrated into a miniaturized and compact system for gas sensing, as shown in Figure 1, maintaining or even enhancing the achievable sensitivity [132]. In this subsection, we introduce the progress and applications of the fully integrated MIR laser spectrometer, focusing on QCL or ICL-coupled HWG gas sensors.

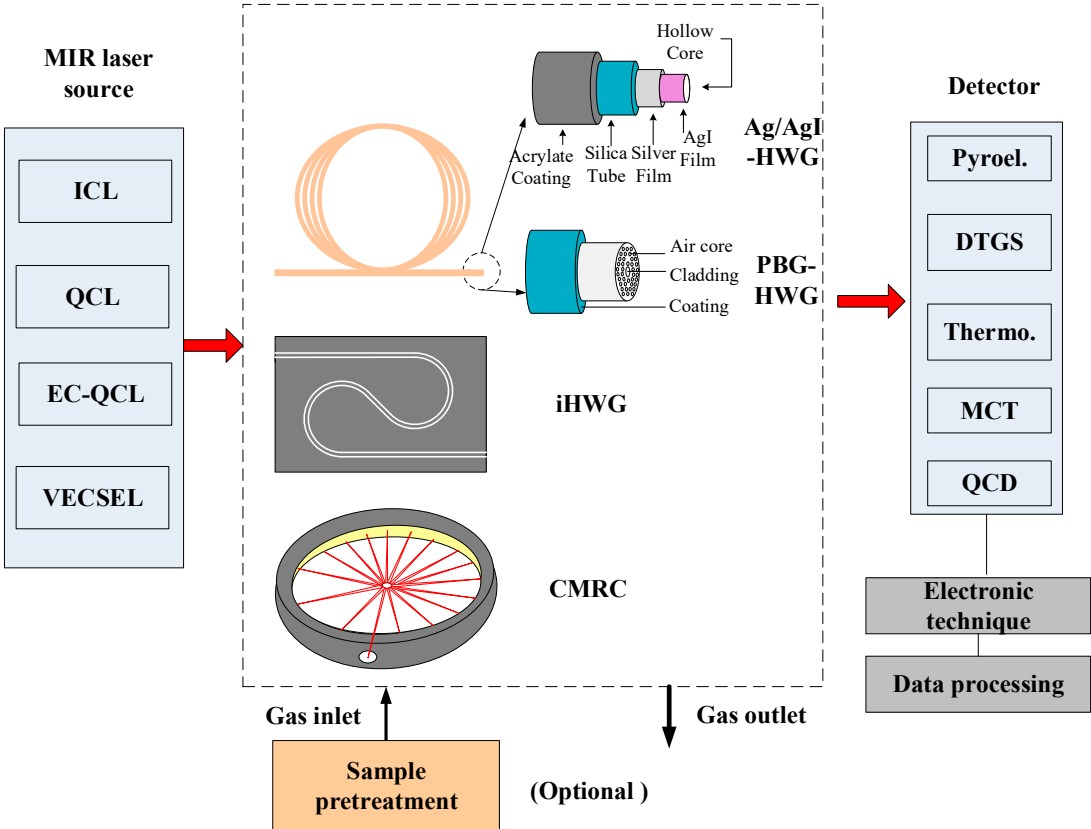

**Figure 1.** Overview of the recently emerging gas cell-based laser gas sensing principles. A gas cell with small volume and easy integration includes: Ag/AgI-HWG-Ag/AgI-coated hollow waveguides, PBG-HWG-photonic bandgaps hollow waveguides, iHWG-substrate-integrated hollow waveguides and CMRC-circular multi-reflection cell. The laser source includes: ICL—interband cascade laser, QCL—quantum cascade laser, EC-QCL—external cavity coupled QCL. The detector includes: Pyroel. —pyroelectric detector, DTGS—deuterated triglycine sulfate detector, Thermo. —thermopile detector, MCT—mercury cadmium telluride semiconductor detector, QCD—quantum cascade detector.

An integrated sensor consists of a compact MIR laser, a new gas cell, and a detector, as shown in Figure 1. The configuration provides an attractive solution for miniaturized and practical applications. The laser should be a QCL, ICL, EC-QCL, or VECSEL. The miniaturized gas cells, i.e., HWG or iHWG, cover the wavelength range of 3.0–11.0 μm, and provide a response time as fast as seconds [32] and a sample volume of sub-milliliter [131]. iHWG-based cells provide excellent modularity and

mechanical stability, with effective OPL of hundreds of millimeters and larger losses per unit length than Ag/AgI-HWG-based cells [32], which could achieve an effective OPL of several meters. The sensor could operate in the DAS, WMS, or intrapulse modulation regime as well [33,133–136].

### 3.5. Open Path Detection without Retroreflectors

The requirements of non-fixed open-path gas detection, e.g., the atmospheric environmental monitoring, leak detection, security early warning, etc., promoted the standoff remote sensing. We focus on the open-path-averaged gas concentrations by the backscatter light from a remote hard target or topographic target. Other open-path systems that are deployed as point samplers or long-path with retroreflectors are beyond the scope of this review.

In order to realize standoff gas detection, a number of laser-based techniques are available, such as TDLAS [34,35,137,138], PAS [139], differential absorption lidar (DIAL) [140], CLaDS [141], and more recently active coherent laser spectrometers (ACLaS) [36]. The basic architectures of these techniques are shown in Figure 2. The OPL is often variable and unknown; therefore, performance is often quoted in a similar fashion to that of open path gas detectors, using the pathlength-integrated unit of ppm·m [142]. The detection limits for such systems commonly range from sub ppm·m to several hundred ppm·m [35,36,141], typically over distances of open path from several meters to hundreds of meters or even kilometers. The performance of this system is typically limited by the level of received laser power, which is dependent on the incident laser power, distance between receiver and scattering target, type of scattering materials, and size of the receiver aperture [142–144]. High-power and broadband tuning lasers, such as OPO and EC-QCLs, are desired to achieve a higher signal-to-noise ratio (SNR) [145–147]. However, high-sensitive and low-noise detectors are particularly important for applications in public areas where eye safety should be considered.

Recently, standoff trace detection has achieved tremendous progress and been applied in leaks [35], environmental monitoring [141], explosives [148,149], combustion [147], unmanned aerial vehicles (UAV) [150,151], and many other promising applications as well.

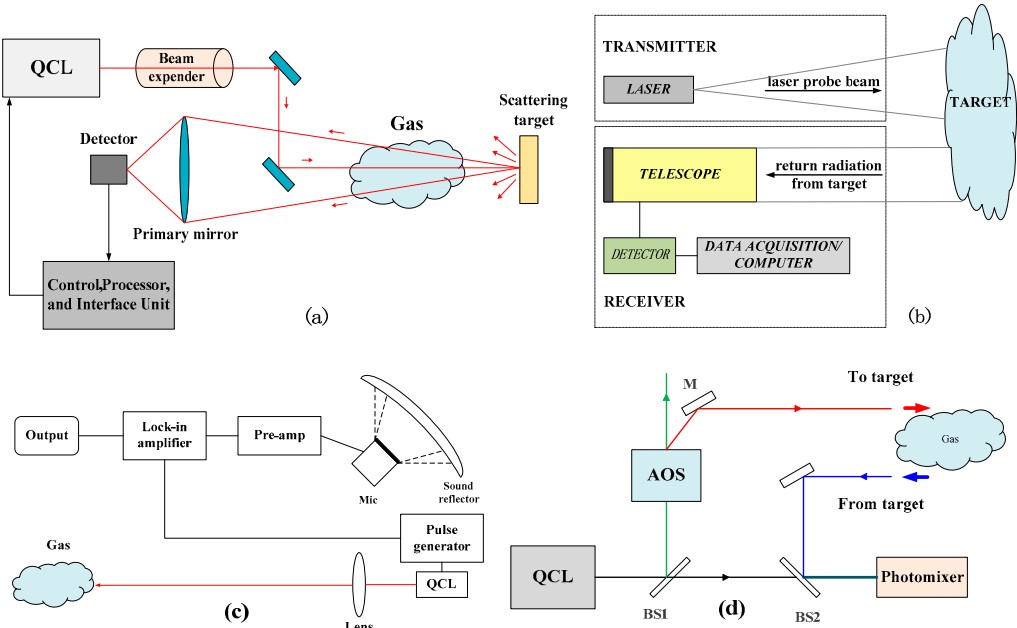

**Figure 2.** Diagrams of the basic architectures of standoff gas detection techniques with non-cooperators: (**a**) TDLAS [35], (**b**) DIAL [149], (**c**) PAS [139], (**d**) ACLaS [36].

## 4. Detection Methods: Derivative Spectroscopy

MIR spectroscopy is attractive for the strong fingerprint signature, and the commonly used DAS often suffers from low selectivity for the spectra overlapping of broadband absorption. Derivative spectroscopy uses first or higher derivatives of absorbance with respect to wavelength for qualitative analysis and for quantification with higher selectivity [152]. The derivatization of zero-order spectrum can lead to separation of overlapped signals or elimination of background caused by other compounds in a sample [153,154].

Derivative spectra could be obtained by the Savitzky-Golay (SG) smoothing/differentiation procedure, which is widely implemented in instrumental software or in packages for spectral data processing. The resolution enhancement in the second derivative spectrum depends on the data spacing in original spectra, absorption peak profile, parameters of SG (i.e., window size and polynomial order). To maximize the separation of the peaks in a second derivative spectrum, the original spectra should be recorded at high resolution and using appropriate parameters [155]. Other methods used to calculate the derivative spectra include numerical differentiation [156] and continuous wavelet transform [157]. The latter has been proven to be efficient in the analysis of overlapping spectra and is advantageous for providing higher SNR and flexibility in searching for absorption peaks.

Derivative spectroscopy can be used in various spectral region including UV [158], visible [159], NIR [160,161], and MIR. The method has a close dependence on instrumental parameters, like speed of scan, the linewidth, and SNR. The derivatization can amplify the noise signals in the resulting curves, which normally leads to a higher SNR. Another disadvantage is the non-robust character of the selected parameters of the elaborated methods. The selected parameters of this method are applicable only for the studied system and every change in composition requires re-optimization and the selection of new parameters of derivatization [153]. Without a homogeneous protocol of optimization, the parameters of the method vary, and most researchers did not describe the parameter selection in their published articles.

## 5. Detection Methods: Modulation Spectroscopy for Wideband Absorption

WMS has been demonstrated to have high sensitivity for sensing gas with an isolated spectral line. When the modulation index is small, WMS is approximately expressed as derivative spectroscopy. For the MIR spectral region, however, the fingerprint spectra are often broad, serried, crowded, and even overlap within the coverage of a tunable laser. To ensure the detection sensitivity and selectivity, the essential procedures include optimizing modulation index, varied modulation amplitude, removing fringes and noise interference, and multicomponent spectral fitting.

### 5.1. Optimizing the Modulation Index

Since the recorded harmonic signals used in WMS are heavily dependent on the spectral line profile and modulation index adopted in the WMS system, the situation for sensing a larger molecular gas with a broadband spectrum is quite different. Moreover, the similarity of signal waveform between a broadband spectrum and the intrinsic optical fringes interference will seriously complicate the signal processing [30] and deteriorate the sensing sensitivity and precision.

The modulation index always plays a pivotal role in WMS-based measurement. A modulation index of 2.2 is recognized as the optimum to achieve the maximum SNR with isolated Gaussian or Lorentzian line profile. For broadband spectrum, however, the reordered harmonic signal would broaden to overlap with the adjacent spectrum, interference, and optical fringes with the so-called modulation index optimum. The overlapping may deteriorate and even disable the WMS measurement. So the modulation index determination should balance the spectra discrimination and the SNR in WMS with broadband spectrum.

We investigated the determination of modulation index for the broadband absorption, taking the carbon disulfide ($CS_2$) spectrum around 2177.6 cm$^{-1}$ as an example. To evaluate the spectra discrimination of the WMS-2f signal, we particularly defined a parameter, SD [27]:

$$SD = \frac{h_{CD}}{0.5 \times (h_A + h_B)},\tag{5}$$

where $A$ and $B$ are the adjacent valleys of neighboring WMS-2f signals, $C$ is the middle point of the line connecting $A$ and $B$, point $D$ at the signal curve is vertical to point $C$, $h_A$ and $h_B$ are the amplitude of absorption valley $A$ and $B$, respectively. $h_{CD}$ is the height difference between point $C$ and $D$. The parameter $SD$ is a constant from 0 to 1, which represents the WMS-2f completely overlapping and parting, respectively, as shown in the insert panel of Figure 3b.

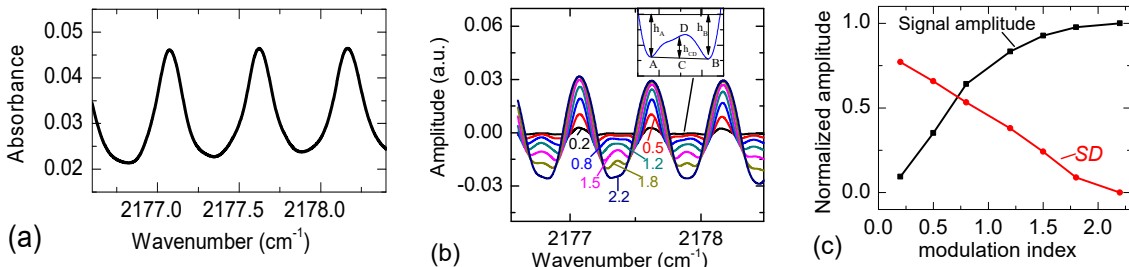

**Figure 3.** (**a**) broadband spectra of $CS_2$ around 2176 cm$^{-1}$ with 30.5 ppm × 5 m; (**b**) simulated WMS-2f signals of Figure 1a with modulation index 0.2 to 2.2, the insert panel describing the definition of spectrum discrimination (SD); (**c**) the SD and normalized WMS-2f signals amplitude under various modulation index [27].

The parameter $SD$ and normalized amplitude of WMS-2f signals under different modulation index was plotted in Figure 3c, which reverse with the modulation index. To balance the $SD$ and normalized amplitude of WMS-2f signals, the modulation index around 1.0 should be a good compromise for optimizing WMS with the band spectrum of $CS_2$ detection.

### 5.2. Varied Modulation Amplitude

A modulation amplitude setting always confronts multi-spectrum with different widths, which creates a significant dilemma. Any single modulation amplitude cannot cover a large spectral width difference, e.g., a difference of the half width at half maximum is more than 50%. A practical way to achieve this is to use a varied modulation amplitude for multi-spectrum detection, namely WMS with varied modulation amplitude (WMS-VMA) [162].

The WMS-VMA was realized mainly by a homemade digital lock-in amplifier (DLIA), which performs the modulation, demodulation of WMS-1f and WMS-2f, and generation of reference signal by an integrated field-programmable gate array-based circuit. The DLIA generates an arbitrary waveform signal by direct digital frequency synthesis. The sine waveform with varied amplitude is prepared in advance and stored in the DLIA's random access memory, and then read out to control the sequence using the frequency controller integrated in the DLIA. The method has been verified by a multiparameter optical sensor with typical broadband spectroscopy [162].

### 5.3. Removing Fringes and Noise Interference

LAS always suffers interference from electric noise and optical fringes; the latter are caused by multiple reflections upon optical interface, i.e., the so-called "etalon effect." The interference causes low sensitivity and precision in the spectrometers. Since the optical fringes are small and sine-like in a well-designed and -fabricated system, the signal profiles of molecular absorption always exhibit distinct differences to those of optical fringes and electric noise. The WMS-2f signal profile would

inherit their difference with an optimized modulation index. So, the molecular absorption can be distinguished from optical fringes and electric noise by the signal profile.

An empirical mode decomposition (EMD) algorithm has been successfully used in WMS to remove optical fringes [27] and noise [163] by characterization of the signal profile. The procedure for employing EMD to decompose and reconstruct the WMS-2f signal is described in Figure 4a. The detected 2f signal was decomposed into intrinsic mode functions (IMF), which have a specific physical source and are meaningful. The simplest criterion, whether IMF from molecular absorption or interference is the correlation coefficient (R) between an IMF and the reference 2f signal, is shown in Figure 4b,c.

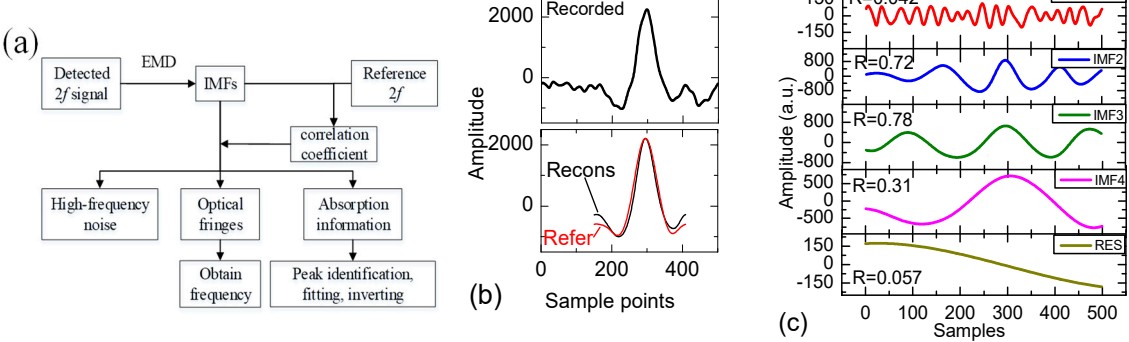

**Figure 4.** Schematic of removing optical fringes and noise with the empirical mode decomposition (EMD) algorithm. (**a**) flow chart of the algorithm; (**b**) WMS-2f signal: recorded in the upper panel, the reconstructed and simulated in lower panel; (**c**) the intrinsic mode functions (IMF) decomposed from the recorded WMS-2f [27].

The components with lower *R* most likely come from fringes or noise, while components with larger *R* with reference absorption must be the absorption. All these components with higher *R* can be added together to reconstruct a WMS-2f free from the interference of fringes and noise. Generally, this method can improve the sensor sensitivity by about 30% [164].

*5.4. Multicomponent Spectral Fitting*

Multicomponent spectral fitting is mainly used to eliminate spectral interference, which commonly occurs in MIR detection. Furthermore, benefitting from the use of multivariate regression and nonlinear least square fitting, it can calculate the multi-component concentration in a gas mixture, i.e., achieving multiple component detection simultaneously using a single DFB ICL [164,165]. The Levenberg-Marquardt (LM) algorithm, also known as the damped least-squares method, was applied for the data fitting. Reference WMS-2f signals of all the components were obtained beforehand. The concentrations of all components in the mixture comply with the constraint condition of non-negative parameters. In each iteration of the fitting routine, the WMS-2f signal of the mixture was simulated with the updated parameters. Once the routine converged, the best fitting parameters were determined as the concentrations of the components.

Multicomponent spectral detection could benefit from the redundancy of the multiple spectra, not only in the magnitude of the absorption but also in the line shape related to temperature and pressure broadening. To make full use of the information buried in the detected spectral lines, we presented an improved multicomponent spectral fitting algorithm for the sensor. We also applied the method of normalizedWMS-2f by 1f for the sensor immunity of laser energy fluctuation [162].

## 6. Detection Methods: Optical Frequency Comb Spectroscopy

Frequency combs enjoy high spectral resolution and broad spectral coverage that make them a unique spectroscopic tool for precision spectroscopy and for multi-species detection [80,166]. Direct



frequency comb spectroscopy (DFCS) employs optical frequency combs to probe spectral features in a parallel fashion [167]. We briefly review the DFCS including frequency comb-based Fourier transform spectroscopy (FC-FTS), cavity-enhanced direct frequency comb spectroscopy (CE-DFCS), and virtual imaging phased array spectroscopy (VIPAS).

*6.1. Frequency Comb Fourier Transform Spectroscopy*

FC-FTS is the measurement of the Fourier transform of the interferogram based on frequency combs source, which offers excellent spectral brightness and spatial coherence. There are two implementations of FC-FTS: Michelson interferometer-based Fourier transform spectroscopy and dual-comb spectroscopy (DCS) [168]. Each presents its own distinct advantages, but both rely on the same physical principle.

**Michelson interferometer-based FC-FTS**. In FC-FTS, OFC works as the light source for Fourier transform spectrometers (FTS). The high spectral brightness, together with the spatial and temporal coherence of the combs, enable acquisition times orders of magnitude shorter than in conventional FTIR spectroscopy. Thus, FC-FTS could be promising in standoff chemical sensing of transient, non-repeatable phenomenal like combustion, plasmas, and explosions [169]. Presently, the applications of FC-FTS is still limited by spectral width, lower comb intensities, mechanically scanned mirrors to record the interferogram, placing limits on the acquisition of spectra [170]. The interferometer records the interference pattern between two combs as slightly different because the light reflected by the moving mirror of the interferometer is Doppler shifted [171,172].

**Dual comb spectroscopy.** DCS uses two frequency combs of slightly differing line spacing, one for reference and the other for sample detection [168]. From each pair of optical lines, one from each comb, a radio frequency beat note is generated on a detector. In this way, optical frequencies are converted into radio frequencies such that the amplitude and phase changes caused by the interaction of one of the combs with a sample can be detected. DCS has an advantage similar to a Michelson interferometer but without moving parts and employing a single point detector, in which case a minimum detectable absorption ~$1 \times 10^{-8}$ cm$^{-1}$ has been demonstrated [173].

Despite an intriguing potential for the measurement of molecular spectra spanning tens of nanometers within tens of microseconds at Doppler-limited resolution, the development of dual-comb spectroscopy is hindered by the demanding stability requirements of the laser combs [168,174,175]. For an ideal, the interference sampled waveform can be Fourier transformed to display the signal spectrum. In reality, the main difficulty comes from the time and phase fluctuations of the frequency comb. However, we experimentally demonstrate that the means of real-time dual-comb spectroscopy can be used to overcome this problem [176]. The true value of DCS for sensitive molecular detection lies in the MIR [177,178], and compact design will be obtained using the semiconductor laser comb technology in the future [179–181].

*6.2. Cavity-Enhanced Direct Frequency Comb Spectroscopy*

CE-DFCS combines broad spectral bandwidth, high spectral resolution, precise frequency calibration, and ultrahigh detection sensitivity all in one experimental platform based on an optical frequency comb interacting with a high-finesse optical cavity [166,182]. Michael et al. demonstrate a minimum detectable absorption of $8 \times 10^{-10}$ cm$^{-1}$, a spectral resolution of 800 MHz, and 200 nm of spectral coverage [183]. Moreover, combined with VIPA technology, the minimum detectable concentration is $1.7 \times 10^{11}$ cm$^{-3}$ [184]. Whereas, a great deal of CE-DFCS applications are in the visible and NIR spectral region, MIR CE-DFCS is attracting a lot of attention [182,185,186]. It is noteworthy that the mode spacing between the cavity and comb should be well matched, or the operation will always too be complicated [186,187].

### 6.3. Virtual Imaging Phased Array Spectroscopy

VIPAS provides alternative approaches that circumvent this problem by directly measuring the power or phase of individual comb teeth that have interacted with the molecular gas [188]. In this case, a novel high-resolution crossed spectral disperser is employed to project the various frequency comb modes onto a two-dimensional digital camera. The distinguishing feature of the present approach is the use of a side-entrance etalon called a VIPA disperser in the visible spectral range. The multiple reflections within the VIPA etalon interfere such that the exiting beam has different frequencies emerging at different angles. Sensing with a broadband comb directly interrogates an absorbing sample, after which the spectrum is dispersed in two dimensions and sensed with detector arrays. Thus, the spectrometer transforms the one-dimensional comb into something more reminiscent of a two-dimensional 'brush'.

With the VIPA method, Diddams et al. [189] resolved 2200 comb modes covering a 6.5 THz span with resolution 1.2 GHz, while Gohle et al. [190] resolved 4000 modes covering a 4 THz span with resolution 1 GHz. However, due to the limitations in optical coating and array detector technology, VIPA is available in the visible and NIR ranges. Proof-of-principle demonstrations have been carried out in the MIR wavelength region spectral resolution 600 MHz ($0.02 \text{ cm}^{-1}$) [191], and resolution 1 GHz ($0.03 \text{ cm}^{-1}$) [192]. MID APD arrays may change this situation in the near future.

## 7. Summary of MIR Gas Sensing

MIR trace gas sensing in the molecular fingerprint region developed rapidly in the near decade mostly due to the commercialization of MIR tunable lasers, i.e., DFB-QCL and DFB-ICL, which could be proven by rough statistics on the number of studies published annually on TOPIC: (Mid-Infrared Lasers) AND TOPIC: (Sensing) in the Web of Science, as shown in Figure 5. Though many prominent works on MIR trace gas sensing were performed before 2008, NO was detected through the $\nu_1$ band near $1875 \text{ cm}^{-1}$ with limit of detection (LOD) of ppb level by DAS [193] and WMS [194], respectively, for industrial processes and vehicle emissions monitoring. We only summarize gas detection based on MIR absorption spectroscopy with tunable lasers in the near decade for the last 10 years; see Table 2 for single component detection and Table 3 for multi-component detection.

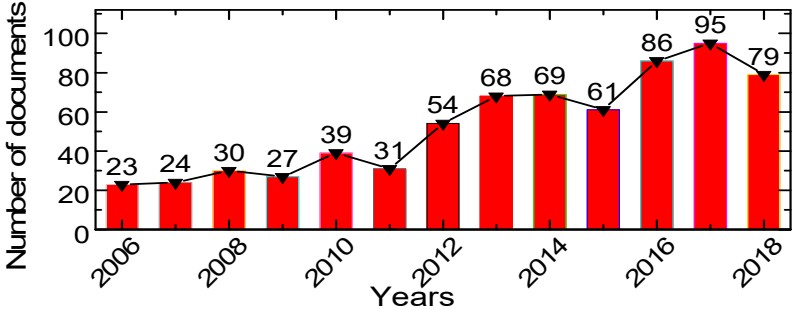

**Figure 5.** Census on mid-infrared lasers AND sensing (TOPIC) in the Web of Science.

**Table 2.** Summary of single component detection by MIR spectroscopy (2008–2018).

| Species | Bands | Wavelength/nm | Laser Type | Techniques | LOD [1]/ppb | Refs | Applications |
|---|---|---|---|---|---|---|---|
| $CH_4$ | $\nu_1$ | 3392 | ICL | WMS-2f | 48@0.1 s | [8] | Exhaled breath analysis |
| | | 3451.9 | DFG | HPSDS [2] | 250 | [195] | Technique research |
| | $\nu_3$ | 3291 | ICL | SA-DAS DAS | 6.3@240 s; 2.25@2.5 s | [196, 197] | Atmospheric |
| | | | | DAS | 1.4@60 s | [198] | |
| | | | | WMS | 13.07@2 s | [199] | |
| | | 3300 | | DAS | 15@60 s | [73] | |
| | | 3240 | | OF-CEAS [3] | 3@2 s | [200] | Technique research |
| | | 3366 | | DAS | $3.8 \times 10^4$ | [63] | Industrial emission and process control |
| | | 3260 | QW-DFB-DL [4] | PAS;WMS | $1.5 \times 10^4$@12 s | [201] | Environmental |
| | | 3200 | DFG | FCS | 60@80 ms | [202] | Atmospheric |
| | | 3250 | DROPO [5] | FCS | 4@15 ms | [203] | |
| | | 3390 | OPO | FCS | / | [204] | Technique research |
| | | 3270 | OPO | NICE-OHMS [6] | 0.09@20 s | [205] | Ultrasensitive detection research |
| | | | GaSb laser | WMS | 13 | [150] | Atmospheric |
| | $\nu_4$ | 7791 | QCL | CLaDS [7] | 60 ppb@100 s | [141] | |
| $C_2H_6$ | $\nu_{10}$ | 3330 | ICL | WMS | 1.5@23 s | [206] | Technique research |
| | | 3340 | | WMS-2f; WMS-2f/1f DAS; | 1.2@4 s; 1.0@4 s; 7.92@1 s | [65, 66] | Atmospheric |
| | | 3360 | LD | WMS | 0.13@1 s | [207] | Environmental |
| | | | LD | WMS-2f | 0.24@1 s | [208] | Technique research |
| $C_2H_4$ | $\nu_1$ | 3266 | ICL | WMS-2f | 53@24 s | [64] | Industrial emission |
| $C_2H_2$ | $\nu_{4+}\nu_5$ | 7263 | EC-QCL | WMS-2f/1f | 3@110 s | [53] | Exhaled breath analysis |
| $C_3H_8$ | $\nu_2$ | 3370.4 | ICL | WMS-2f | 460@1 s | [209] | Leakage monitoring |
| $C_5H_8$ | / | 3333.3 | OPO | FCS | 7@30 s | [210] | Technique research |
| $C_6H_6$ | $\nu_{14}$ | 9640 | QCL | DAS | 12@200 s | [211] | Atmospheric |
| $C_{10}H_{22}$ | / | 3380 | ICL | PAS | 0.3 | [212] | Industrial process |
| $H_2CO$ | $\nu_4$ | 3493 | ICL | DAS | $10^3$ | [213] | Workplace monitoring |
| | | 3356 | | WMS-2f | 73@40 s | [214] | Combustion emission |
| | $\nu_1$ | 3599 | ICL | DAS; WMS | 0.26@300 s; 0.069@90 s | [215] | Technique research |
| | | | | WMS-2f | 1.5@140 s | [216] | Atmospheric |
| | | | | DF-RFM [8] | 25@1 s | [67] | Technique research |

**Table 2.** *Cont.*

| Species | Bands | Wavelength/nm | Laser Type | Techniques | LOD [1]/ppb | Refs | Applications |
|---|---|---|---|---|---|---|---|
| $CH_3OH$ | $\nu_9$ | 3390 | OPO | FCS | 40@30 s | [210] | Technique research |
| $C_2H_5OH$ | / | 3367 | OPO | FCS | 40@30 s | [210] | Technique research |
| $CH_3COCH_3$ | $\nu_9$ | 3380 | VECSEL [9] | DAS | 13@300 s | [217] | Breath VOCs detection |
| | | 3389.8 | OPO | FCS | 9.1@30 s | [210] | Technique research |
| | $\nu_5$ | 8000 | ECQCL | WMS-2f | 15@10 s | [52] | Spoilage monitoring of agricultural products |
| | $\nu_1$ | 3300 | OPO | DIAL [10] | $1.2 \times 10^5$ | [140] | Atmospheric |
| $CO_2$ | $\nu_3$ | 4330 | QCL | I-QEPAS [11] | 300 ppt@4 s | [218] | Technique research |
| | | 4200 | ICL | DAS | $5 \times 10^4$ | [72] | Combustion diagnose |
| | | 4172 | | | / | [219] | |
| CO | $\nu_1$ | 4691.2 | ICL | WMS | 9@0.07 s | [71] | Exhaled breath analysis |
| | | 4600 | | DAS | 500@14 s | [220] | Atmospheric |
| | | 4764 | QCL | WMS | 26@1 s | [221] | Indoor air |
| | | 4980 | | WMS | / | [222] | Combustion diagnose |
| NO | $\nu_1$ | 5184 | ICL | DAS | $3 \times 10^4$@10 ms | [223] | Combustion emission |
| | | 5200 | | QEPAS [12] | 120 | [224] | Engine exhaust monitoring |
| | | 5250 | QCL | WMS-2f | / | [225] | Technique research |
| | | 5030 | | DAS | / | [226] | Gas sensing in high temperature |
| | | 5263 | | I-QEPAS; WMS-2f | 4.8@30 ms | [227] | Environmental |
| $NO_2$ | $\nu_3$ | 6250 | QCL | WMS | 360 (600 K); 760 (800 K) | [228] | Gas sensing in high temperature |
| | $\nu_1 + \nu_3$ | 3250–3550 | OPO | PAS | 14@170 s | [3] | Environmental pollution |
| $N_2O$ | $\nu_3$ | 7782 | ECQCL | DAS | $7.36 \times 10^3$ | [229] | Toxic industrial chemical detection |
| | $\nu_2$ | 8600 | QCL/DFG | FCS | 0.3 | [230] | Atmospheric |
| | $\nu_1$ | 4530 | QCL | CLaDS | $1.2 \times 10^3$ | [231] | Atmospheric |
| $NH_3$ | $\nu_2$ | 10,400 | ECQCL | PAS | 1 | [49] | Atmospheric |
| | | 10,340 | QCL | QEPAS; 2f-WMS | 6@1 s | [232] | Exhaled breath analysis |
| | | 9560 | | DAS | 17.3@3 s | [233] | Atmospheric |
| | | 9060 | | WMS | 0.3 | [2] | Atmospheric |
| | $\nu_1$ | 2958.5 | OPO | FCS | 25@30 s | [210] | Exhaled breath analysis |
| $N_2H_4$ | $\nu_{12}$ | 10,363 | LD | DAS | 400 | [234] | Chemical analysis |
| $O_3$ | $\nu_3$ | 9697 | QCL | DIAS [13] | 300 | [235] | Atmospheric |

**Table 2.** *Cont.*

| Species | Bands | Wavelength/nm | Laser Type | Techniques | LOD [1]/ppb | Refs | Applications |
|---|---|---|---|---|---|---|---|
| $H_2O$ | $\nu_2$ | 6700 | QCL | OA-ICOS [14] | 280 | [21] | Chemical analysis |
| | $\nu_3$ | 2666.7 | OPO | FCS | 5.3@30 s | [210] | Technique research |
| $H_2O_2$ | $\nu_6$ | 7730 | QCL | QEPAS | 12@100 s | [236] | Breath diagnosis |
| | $\nu_3$ | 3760 | OPO | FCS | 8 | [171] | Exhaled breath analysis |
| $H_2S$ | $\nu_2$ | 7900 | QCL | QEPAS | 450@3 s 330@30 s | [237] | Environmental pollution |
| $SF_6$ | $\nu_3$ | 10,540 | QCL | QEPAS | 0.05@1 s | [238, 239] | Technique research |
| $CS_2$ | $\nu_1 + \nu_3$ | 4590 | QCL | QEPAS | 28@1 s | [240] | Industrial process |
| | | | | DOAS; WMS | 10.5; 60@240 s | [27, 241] | Atmospheric |
| OCS | $\nu_3$ | 4860 | QCL | DAS | 1.2@0.4 s | [242] | Exhaled breath analysis |
| $CH_3SH$ | $\nu_2$ | 3393 | ICL | DAS | 25@1.84 s | [68] | Industrial emission |
| | | 3392 | | WMS | 7.1@295 s | [165, 243] | Atmospheric |
| $CH_3SCH_3$ | $\nu_{18}$ | 3370 | ICL | WMS-2f/1f | 2.8@125 s | [69] | Environmental |
| | | 3337 | ICL | WMS | 9.6@164 s | [164] | Atmospheric |

[1] LOD: Limit of Detection; [2] HPSDS: Heterodyne phase sensitive dispersion spectroscopy; [3] OF-CEAS: Optical feedback cavity-enhanced absorption spectroscopy; [4] QW-DFB-DL: Quantum wells distributed feedback diode laser; [5] DROPO: Doubly resonant optical parametric oscillator; [6] NICE-OHMS: Noise-immune cavity-enhanced optical heterodyne molecular spectrometry; [7] CLaDS: Chirped laser dispersion spectroscopy; [8] DF-RFM: Dual-feedback RF modulation; [9] VECSEL: Vertical-external cavity surface-emitting laser; [10] DIAL: Differential absorption lidar; [11] I-QEPAS: Intracavity Quartz-Enhanced Photoacoustic Spectroscopy; [12] QEPAS: Quartz-enhanced photoacoustic spectroscopy; [13] DIAS: Differential absorption spectroscopy; [14] OA-ICOS: Off-axis integrated cavity output spectroscopy;

**Table 3.** Summary of information on multi-component simultaneous detection in the last 10 years.

| Species | Bands | Wavelength/nm | Laser Type | Techniques | LOD/ppb | Refs | Applications |
|---|---|---|---|---|---|---|---|
| $CH_4/C_2H_6$ | $\nu_3/\nu_{10}$ | 3291 | ICL | DAS | 5@1 s | [244] | Atmospheric |
| | | 3337 | | | 8@1 s | | |
| $CH_4/C_2H_6$ | $\nu_3/\nu_{10}$ | 3291 | ICL | DAS | 2.7@1 s | [245] | Atmospheric |
| | | 3337 | | WMS-2f | 2.6@3.4 s | | |
| $CH_4/C_2H_6$ | $\nu_3/\nu_{10}$ | 3337 | ICL | WMS | 17.4@4.6 s | [246,247] | Atmospheric |
| | | | | | 2.4@4.6 s | | |
| $CH_4/C_2H_6$ | $\nu_3/\nu_{10}$ | 3404 | DFG | CLaDS | 360@1 s; 60@100 s | [248] | Technique research |
| | | 3335.5 | | | / | | |
| $CH_4/C_2H_6/C_3H_8$ | $\nu_3/\nu_{10}/\nu_1$ | 3345 | ICL | QEPAS | 90@1 s | [249] | Oil and gas industry monitoring |
| | | | | | 7@1 s | | |
| | | | | | $3 \times 10^3$@1 s | | |
| $CH_4/CO/H_2CO$ | $\nu_4/\nu_1/\nu_2$ | 7880/4633 5683 | QCL | WMS-2f | 0.5 for $H_2$ CO@2 s | [250] | Atmospheric |
| $CH_4/N_2O$ | $\nu_4/\nu_3$ | 7700 | QCL | DOAS | $3 \times 10^4$ | [35] | Remote gas leakage detection |
| | | | | | $3.3 \times 10^3$ | | |

**Table 3.** *Cont.*

| Species | Bands | Wavelength/nm | Laser Type | Techniques | LOD/ppb | Refs | Applications |
|---|---|---|---|---|---|---|---|
| $CH_4/N_2O$ | $\nu_4/\nu_3$ | 7800 | QCL | WMS-2f | 5.9@1 s<br>2.6@1 s | [114] | Atmospheric |
| $CH_4/H_2CO/C_2H_4/$ $C_2H_2/CO$ | $\nu_3/\nu_1/\nu_{11}/\nu_3/\nu_1$ | 2500–5000 | OPO | FCS | 1.7<br>310<br>320<br>110<br>270 | [251] | Technique research |
| $CH_4/NO$ | $\nu_3/\nu_1$ | 3000–5400 | DROPO | FCS | 20<br>15 | [172] | Technique research |
| $CH_4/N_2O/H_2O$ | $\nu_4/\nu_3/\nu_2$ | 7710 | QCL | DAS | 23@1 s<br>6.5@1 s<br>$6.2 \times 10^4$@1 s | [252] | Atmospheric |
| $CH_4/N_2O/H_2O$ | $\nu_4/\nu_3/\nu_2$ | 8000 | ECQCL | WMS-2f | 4.8@1 s<br>0.9@1 s<br>$3.1 \times 10^4$@1 s | [50] | Atmospheric |
| $CH_3OH/C_2H_5OH$ | $\nu_8$ | 10,100 | ECQCL | DAS | 130@1 s<br>$1.2 \times 10^3$@1 s | [146] | Atmospheric VOCs |
| $C_2H_5OH/(C_2H_5)_2O/$ $CH_3COCH_3$ | / | 3800 | QCL | CRDS | 157<br>60<br>280 | [18] | Atmospheric VOCs |
| $CO_2/CO$ | $\nu_3/\nu_1$ | 4730 | EC-QCL | WMS-2f | $6.5 \times 10^5$<br>9 | [7] | Exhaled breath analysis |
| $CO_2/CO$ | $\nu_3/\nu_1$ | 4250<br>4860 | QCL | WMS | $10^6$ | [51] | Combustion diagnose |
| $CO_2/CO$ | $\nu_3/\nu_1$ | 4193<br>4979 | ICL<br>QCL | DAS | / | [253] | Combustion diagnose |
| $CO_2/N_2O$ | $\nu_3/\nu_1$ | 4466 | QCL | DAS<br>WMS | 2.7; 0.2<br>4.3@1 s;/ | [118] | Technique research |
| $H_2O/CO_2/CO$ | $\nu_3/\nu_3/\nu_1$ | 2551<br>4176<br>4865 | LD<br>ICL<br>QCL | WMS-2f/1f | $1.4 \times 10^7$<br>$6 \times 10^6$<br>$4 \times 10^6$ | [147] | Combustion diagnose |
| $NO/CO/$ $N_2O$ | $\nu_1$ | 5263<br>4566 | QCL | DAS | 0.5@1 s<br>0.8@1 s | [254] | Exhaled breath analysis |
| $CO/N_2O$ | $\nu_1$ | 4500 | QCL | WMS | 0.36<br>0.15 | [255] | Atmospheric |
| $CO/N_2O$ | $\nu_1$ | 4610 | QCL | QEPAS | 0.09@5 s<br>0.05@5 s | [256] | Atmospheric |
| $NO/NO_2$ | $\nu_1/\nu_3$ | 5263<br>6135 | QCL | DAS | 597.3@1 s<br>438.3@1 s | [257] | Atmospheric |
| $NO/NO_2$ | $\nu_1/\nu_3$ | 5263<br>6134 | QCL | FMS | 4@1 s<br>9@1 s | [258] | Technique research |
| $NO/NO_2$ | $\nu_1/\nu_3$ | 5263<br>6134 | QCL | WMDM-2f [1] | 0.75@100 s<br>0.9@200 s | [259] | Atmospheric |
| $NO/NO_2$ | $\nu_1/\nu_3$ | 5263<br>6250 | QCL | DAS | 1.5@100 s<br>0.5@100 s | [260] | Vehicle exhaust emission |
| $NO/NO_2$ | $\nu_1/\nu_3$ | 5250<br>6250 | QCL | DAS | 1.5@100 s<br>0.5@100 s | [261] | Environmental monitoring |

**Table 3.** *Cont.*

| Species | Bands | Wavelength/nm | Laser Type | Techniques | LOD/ppb | Refs | Applications |
|---|---|---|---|---|---|---|---|
| $NO/NO_2/NH_3$ | $\nu_1/\nu_1/\nu_3$ | 5263 | QCL | WMS | 0.2@100 s; 0.96@1 s | [262,263] | Industrial emission |
|  |  | 6250 |  |  | 0.12@100 s; 0.94@1 s |  |  |
|  |  | 9063 |  |  | 0.1@100 s; 0.86@1 s |  |  |
| $NH_3/C_4H_{10}$ | $\nu_2/\nu_{24}$ | 8500 | QCL | MHS [2] | / | [48] | Technique research |
| $NO_2/$ HONO | $\nu_1/$ | 6234 | QCL | TILDAS [3] | 0.03 | [264] | Atmospheric pollution |
|  |  | 6024 |  |  | 0.3 |  |  |
| $SO_2/SO_3$ | $\nu_3$ | 7500 | QCL | DAS | $(1\text{–}2) \times 10^3$ | [265] | Industrial emission |
|  |  | 7160 |  |  |  |  |  |
| $C_2H_5Cl/CH_2Cl_2/$ $CHCl_3$ | $//\nu_4$ | 7949 | EC-QCL | DAS | 4 | [266] | Atmospheric VOCs |
|  |  |  |  |  | $7 \times 10^3$ |  |  |
|  |  |  |  |  | 11 |  |  |

[1] WMDM-2f: Wavelength modulation-division multiplexing-2f; [2] MHS: Multiheterodyne spectroscopy; [3] TILDAS: Tunable infrared laser differential absorption spectroscopy.

## 8. Conclusions and Future Prospects

MIR spectral trace gas sensing is particularly attractive for its unique and strong fingerprint absorption. Higher sensitivity has been achieved by solving all the challenges of the spectral feature, i.e., broad, serried, crowding, and even overlapping in the MIR region. Benefiting from the methods mentioned above, multicomponent simultaneous detection is an expected achievement.

The invention and commercialization of high-performance MIR lasers, i.e., DFB-QCL and DFB-ICL, promoted the development and application of MIR tunable laser-based trace gas sensors, especially in miniaturized and portable applications. Though more than 100 types of gas have been detected by tunable laser-based sensors, there are huge demands for higher detection sensitivity, or, in extreme conditions or scientific exploration, more other types of gas need to be detected. Thus, we believe greater progress will be achieved in the next decade, which may include:

**(1) Compact integrated gas sensors**. Compact sensors with lower power consumption or battery power could benefit from VCSEL, DFB-ICL, and iHWG. Even a sensor system on a chip will become possible using integrated optics with the fabrication of miniaturized devices integrating the electronics and optics. These sensors could play a vital role in portable and wearable applications that could be applied for breath analysis, diagnostics, metabolomics, environmental safety detection, and related applications.

**(2) Multicomponent gas sensors**. Multicomponent sensors will achieve more progress, benefiting from the wider wavelength coverage by integrated laser arrays, OFC, or EC-QCL. More species could be detected simultaneously by a particularly devised broadband laser, which could expand their applications in scientific research, including combustion diagnosis, chemical reaction process dynamics, exhaled breath analysis, and metabolomics.

**(3) Standoff remote sensing**. The techniques of open-path standoff detection by backscattered MIR light provide a promising method of prompt and flexible assessment of atmospheric environmental, leaks, explosive, and security in handheld devices or UAV. The detection sensitivity could be substantially improved by newly developed high-performance MIR detectors and progress in high-power DFB-QCL.

**(4) Ultra-sensitive sensing**. With the development of a mid-infrared laser source and high-performance detector, combined with cavity enhancement technology and noise immunity technology, ultra-high detection sensitivity becomes possible.

**Author Contributions:** This review article was jointly written and proof-read by all authors. Z.D. proposed the idea, drafted the outline and structure, and contributed to the principle, detection methods as well as conclusion. S.Z. collected references, composed the whole manuscript, and contributed to the summary. J.L. contributed to the system configuration and detection methods. N.G. contributed to the system configuration. K.T. contributed to the detection methods.

**Funding:** This research was funded by the National Natural Science Foundation of China (61505142), the Tianjin Natural Science Foundation (16JCQNJC02100), the Science & Technology Development Fund of the Tianjin Education Commission for Higher Education (2017KJ085), and the Natural Science Foundation of Hebei Province (F2014202065).

**Acknowledgments:** This work was supported by the open project of Key Laboratory of Micro Opto-electro Mechanical System Technology, Tianjin University, Ministry of Education.

**Conflicts of Interest:** The authors declare no conflict of interest.

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
