# Peer review of "Mid-Infrared Tunable Laser-Based Broadband Fingerprint Absorption Spectroscopy for Trace Gas Sensing: A Review"

_applsci, doi:10.3390/app9020338_

Round 1
Reviewer 1 Report
See attached

Author Response
General comments:
1.Table 2/Table 3: Most of the MIR spectrometers included in this table use a single frequency MIR laser source (e.g. DFB-QCL), which do not belong in a review paper on broadband spectroscopy.
Reply: What this manuscript focused exactly is broadband spectroscopy by using single frequency MIR laser source, as descripted in the Line 121 to 127, and Section 4 to 6.
2.Section 3.5: This section deviates significantly from the remit the manuscript. Most of the references describe studies using DFB or other narrowband laser systems. The authors should focus only on the advantages, disadvantages, and demonstrations of broadband standoff detection.
Reply: We do believe that the optical configuration of open path detection without retroreflectors is one of the most important directions for tunable laser absorption spectroscopy recently, and should be mentioned specially in this manuscript.
3.Section 5.4: Multicomponent spectral fitting is a general analytical method that is commonly used with a range of spectral techniques, not just wavelength modulation spectroscopy.
Reply: It has been revised from L492 to L496 P14. Multicomponent spectral fitting is mainly used to eliminate spectral interference which occurs in MIR detection commonly, while achieving multi-component detection by WMS is an unexpected result.
4.References: Several of the references have no relation to the text where they are cited. I would highly recommend a more thorough check of the references by the authors.
Reply: We appreciate your so careful review. We have deleted several references which are cited inappropriately, and we have a more thorough double check of all the references.
5.Sections 4 – 6: The authors spend, in this referee’s opinion, a disproportionate amount of space discussing wavelength modulation spectroscopy, going into extensive detail on the methods being developed to improve data analysis. In contrast, there is a lot of work on mid-infrared frequency comb spectroscopy – particularly to overcome stability requirements of dual frequency comb work – for trace gas sensing that is overlooked (e.g. cavity-locking methods, Vernier spectrometers – see Adler et al. 2010 in Annu. Rev. Anal. Chem. For example)
Reply: Thank you very much for the comments. Wavelength modulation spectroscopy covers most of our work. We have revised and added some content in Dual comb spectroscopy of Section6.1 from L536 to L545 P15.
Specific comments:
1.P1 L57: Reference 2 (Wojtas et al. 2017) is a paper on the PAS technique, and does not include any urban or industrial emissions measurements.
Reply: This reference was cited inappropriately and we have deleted it form L57 P1.
2.P1 L58: It is unclear how Reference 7 (Manninen et al.2012) – which is a manuscript about progress in multipass cell research – relates to chemical analysis and industrial process control.
Reply: This reference was cited inappropriately and we have deleted it form L58 P1.
3.P1 L58: Reference 10 (Li et al. 2017) relates to HWG, and has no relation to actual medical diagnostics measurements.
Reply: This reference was cited inappropriately and we have deleted it form L58 P1.
4.P3 L100-103: I would argue that absorption spectroscopy is not good for quantification of VOCs – the lines are so dense for molecules with more than ~2 carbon atoms that it is difficult to resolve features for specific identification and sensitive detection.
Reply: It’s true that the fingerprint absorption lines are so dense for VOCs molecules in MIR, which results in that it is more difficult to measure these large molecules quantitatively using absorption spectroscopy than to detect small molecules with obvious line absorption feature. But we believe that specific identification and sensitive detection can be achieved with appropriate methods. And we have added a book reference [23] in L103 P3, which covers the relevant topics.
5.P3 L103: “polyatomic molecules” is redundant – all molecules are polyatomic.
Reply: That's a mistake we made. What we want to express is larger organic molecules, and we have replaced “polyatomic molecules” with “larger molecules” in L103 P3, L103 P3, L114 P3 and L119 P3.
6.P3 L131: L is not a dependent variable here – you define it as a path length.
Reply: We have revised formula (1) by deleting L from dependent variable in L131 P3.
7.P4 L151: Please define a, μ, and θ.
Reply: The μ was a slip of the pen, which should be “ ”. We have revised it in L151 P4 and defined “a” as “absorption coefficient”, and “θ” as “phase angle” in L152 P4.
8.P1 L63 and P5 L167-169: “LGA” is not a commonly used acronym in the spectroscopy community. It is used for a few commercial products, but I don’t think an unusual acronym belongs in a review manuscript.
Reply: We accept your suggestion. And we have replaced LGA with LAS in L63 P1, L168-171 P5, L185 P5 and L468 P13.
9.P5 L195: I would be careful about referring to supercontinuum sources as “MIR supercontinuum laser”. While they are laser-pumped, the nonlinear processes result in incoherent output light so it should be made more clear that they are light sources.
Reply: “MIR supercontinuum laser” referring to supercontinuum sources is inappropriate, so we have deleted it from L 195 P5.
10.P5 L195: superluminescent light emitting diodes are not lasers and do not belong in the review.
Reply: “Superluminescent light emitting diodes” are not lasers, so we have deleted it from L 195 P5.
11.P7 L279-280: Please provide references for ICLs, quantum dot, and quantum dash FM combs – all references listed are for QCLs. It may be useful to also to into more detail on quantum dot and dash technology.
Reply: We have readjusted the references[91-97] for QCLs and added a new reference [85] for ICLs, [98] for quantum dot, and [99] for quantum dash respectively in L280-281 P7.
12.P8 L322: It is unclear what “iHWG” stands for.
Reply: Related information which explains what “iHWG” stands for is in L189 P5.
13.Table 1: There is no reason why techniques that do not have the advantage of (e) “suitable for MIR” to be included.
Reply: This was a slip of the pen and we have revised it in Table 1.
14.P10 L369: Reference 137 (Wang et al. 2015) describes research with near-infrared lasers, and therefore does not belong in this discussion.
Reply: This reference was cited inappropriately and we have deleted it form L373 P10.
15.P11 L401-402: Discussion of derivative spectroscopy in the UV to near-infrared spectral regions is not within the scope of this review (References 152 – 155).
Reply: Derivative spectroscopy is a common technique which can be used for the whole spectral range, although the applications of derivative spectroscopy in MIR is focused here, we think that it is better to mention its applications in the UV to near-infrared spectral regions for a better understanding to some readers.
16.P21 L632: While an “ant’s breathing” is a cute image, ants do not have lungs and breath through multiple points on their body.
Reply: An “ant’s breathing” is an image description to show how sensitive that the spectroscopic gas sensing system can achieve in the future. As the reviewer said, ants have no lungs and cannot breathe like mammals. Therefore, from a rigorous point of view, we have deleted that sentence in L637 P21.
17.Reference 198: This refers to work with a diode laser, not a QCL.
Reply: We have checked this reference and revised the mistake out of our carelessness in Table 2.

Reviewer 2 Report
In this manuscript, the authors review recent advances in absorption spectroscopy in the mid-infrared (MIR) region aimed at trace gas sensing. The authors pick up some state-of-the-art tunable or broadband lasers operating in the MIR region, efficient IR-sensitive detectors, and some important methodological approaches in the field of absorption spectroscopy. The list of published papers relating to gas detection based on MIR absorption spectroscopy is comprehensive and it will be greatly helpful for readers. This manuscript will be accepted for publication in Applied Sciences with a revision considering the following comments.
1. In the Introduction section, the authors mention some typical approaches of cavity-enhanced absorption spectroscopy (CEAS) in the line 88-92. However, published papers about the each approach are missing in the reference list. Although the authors show [14] and [15] as reference literatures of CEAS in the reference list, they should add some recent papers about the "each" approach shown in the line 88-92 in addition to [14] and [15].
2. The authors divide one topic: "Detection methods" into three sections from Section 4 to Section 6. The authors should put these sections into one section, i.e. Section 4: Detection methods, Subsection 4.1: Derivative spectroscopy, Subsection 4.2: Modulation spectroscopy for wideband absorption, and Subsection 4.3: Optical frequency comb spectroscopy. Use subsubsections as well, e.g. 4.2.1 Optimizing the modulation index etc.
3. The second table in Table 1 protrudes into the margin because it is too large. The authors should revise it to fit it in the frame. The authors might as well use footnotes in the same way as Table 3.
4. Figure captions should be justified, not centered in Fig. 1 and Fig. 2. In the caption of Fig. 1-4 , use "Figure" instead of "Fig."
5. Type a space between numbers and units, e.g. see some words in the lines from 560 to 564, use "1.2 GHz" instead of "1.2GHz."
6. In some items in Referent section, e.g. . 46, 132, 148, 151 and 222, use a DOI number instead of URL.
Author Response
Comments:
1. In the Introduction section, the authors mention some typical approaches of cavity-enhanced absorption spectroscopy (CEAS) in the line 88-92. However, published papers about the each approach are missing in the reference list. Although the authors show [14] and [15] as reference literatures of CEAS in the reference list, they should add some recent papers about the "each" approach shown in the line 88-92 in addition to [14] and [15].
Reply: Thanks. We have added five new references from [18] to [22] for "each" approach of cavity-enhanced absorption spectroscopy (CEAS) shown in the L 88-92 P3-4.
2. The authors divide one topic: "Detection methods" into three sections from Section 4 to Section 6. The authors should put these sections into one section, i.e. Section 4: Detection methods, Subsection 4.1: Derivative spectroscopy, Subsection 4.2: Modulation spectroscopy for wideband absorption, and Subsection 4.3: Optical frequency comb spectroscopy. Use subsubsections as well, e.g. 4.2.1 Optimizing the modulation index etc.
Reply: Thanks. The suggestions of structure revision are reasonable and logical. However, the intention of the present structure is to emphasize the detection methods, which is the major focus of the manuscript. We are willing to revise it if the reviewer insists.
3. The second table in Table 1 protrudes into the margin because it is too large. The authors should revise it to fit it in the frame. The authors might as well use footnotes in the same way as Table 3.
Reply: We have revised second table in Table 1 to fit it in the frame.
4. Figure captions should be justified, not centered in Fig. 1 and Fig. 2. In the caption of Fig. 1-4 , use "Figure" instead of "Fig."
Reply: We have adjusted the figure captions of Figure. 1 in L359 P10 and Figure. 2 in L387 P11, and revised “Fig” as "Figure" in the caption of Figure. 1-4 in L353 P10, L386 P11, L446 P13 and L483P14.
5. Type a space between numbers and units, e.g. see some words in the lines from 560 to 564, use "1.2 GHz" instead of "1.2GHz."
Reply: We have revised the mistakes from L566 to L568 P16, and checked through the whole manuscript.
6. In some items in Referent section, e.g. . 46, 132, 148, 151 and 222, use a DOI number instead of URL.
Reply: We have checked the “References” section and revised the wrong information with EndNote.

Reviewer 3 Report
The report is very comprehensive and offers an in-depth review of trace gas sensing utilizing mid-infrared lights. That said, the work is light on discussion of current state of the art for infrared detectors as an essential part of the trace gas sensing system. Since it is a review work for this field, it has to include the following references in the infrared detector section for it to be acceptable for publication:
Page 8 - line 303. On hetero-structure based detectors: The authors claim that the fabrication methods makes them costly and difficult for commercial use. That is a true challenge, however, recent advances in the field, has made it cost-effective for fabrication. Specifically, the authors have to cites the following two articles as scalable methods for fabrication of commercially available heterostructure detectors:
P. Dianat, "A Scalable Low-cost Manufacturing to Hybridize Infrared Detectors with Si read-out Circuits," 2018 IEEE Research and Applications of Photonics In Defense Conference (RAPID), Miramar Beach, FL, 2018, pp. 1-3.
doi: 10.1109/RAPID.2018.8508935
M. Zamiri et al., “Antimonide-based membranes synthesis integration and strain engineering,” Proc. Nat. Acad. Sci. USA, vol. 114, no. 1, pp. E1–E8, 2017.
There are minor grammatical mistakes and typos across the article. Given that it is a long review article, it is expected. However, the authors may benefit from an in-depth language review for the paper.
In my opinion, this is an in-depth review that not only is into the interest of researchers in trace-gas sensors, but also other researchers in the fields of infrared detection and sourcing would benefit from. My recommendation is acceptance after major revisions.
Author Response
Comments:
Page 8 - line 303. On hetero-structure based detectors: The authors claim that the fabrication methods makes them costly and difficult for commercial use. That is a true challenge, however, recent advances in the field, has made it cost-effective for fabrication. Specifically, the authors have to cites the following two articles as scalable methods for fabrication of commercially available heterostructure detectors:
P. Dianat, "A Scalable Low-cost Manufacturing to Hybridize Infrared Detectors with Si read-out Circuits," 2018 IEEE Research and Applications of Photonics In Defense Conference (RAPID), Miramar Beach, FL, 2018, pp. 1-3.
doi: 10.1109/RAPID.2018.8508935
M. Zamiri et al., “Antimonide-based membranes synthesis integration and strain engineering,” Proc. Nat. Acad. Sci. USA, vol. 114, no. 1, pp. E1–E8, 2017.
Reply: Thanks. We have polished this section from in L306-309 P8, and cited the recommended two articles (reference [107] and [108]) as scalable methods for fabrication of commercially available heterostructure detectors.

Round 2
Reviewer 3 Report
The authors have resolved my previous concerns in the revised manuscript. I recommend publish as is.
Author Response
Thank you very much for your work and comments.